# Impact of Sociodemographic, Premorbid, and Injury-Related Factors on Patient-Reported Outcome Trajectories after Traumatic Brain Injury (TBI)

**DOI:** 10.3390/jcm12062246

**Published:** 2023-03-14

**Authors:** Nicole von Steinbuechel, Stefanie Hahm, Holger Muehlan, Juan Carlos Arango-Lasprilla, Fabian Bockhop, Amra Covic, Silke Schmidt, Ewout W. Steyerberg, Andrew I. R. Maas, David Menon, Nada Andelic, Marina Zeldovich

**Affiliations:** 1Institute of Medical Psychology and Medical Sociology, University Medical Center Goettingen, Waldweg 37A, 37073 Goettingen, Germany; 2Department Health & Prevention, Institute of Psychology, University of Greifswald, Robert-Blum-Str. 13, 17489 Greifswald, Germany; 3Departments of Psychology and Physical Medicine and Rehabilitation, Virginia Commonwealth University, 907 Floyd Ave., Richmond, VA 23284, USA; 4Department of Biomedical Data Sciences, Leiden University Medical Center, 2333 RC Leiden, The Netherlands; 5Department of Neurosurgery, Antwerp University Hospital and University of Antwerp, 2650 Edegem, Belgium; 6Division of Anaesthesia, University of Cambridge/Addenbrooke’s Hospital, Box 157, Cambridge CB2 0QQ, UK; 7Department of Physical Medicine and Rehabilitation, Oslo University Hospital, 0450 Oslo, Norway; 8Institute of Health and Society, Research Centre for Habilitation and Rehabilitation Models, Faculty of Medicine, Univeristy of Oslo, 0373 Oslo, Norway

**Keywords:** traumatic brain injury, recuperation trajectories, long-term outcomes, health-related quality of life (HRQoL), mental health, Multivariate Latent Class Mixed Models (MLCMM)

## Abstract

Traumatic brain injury (TBI) remains one of the leading causes of death and disability worldwide. To better understand its impact on various outcome domains, this study pursues the following: (1) longitudinal outcome assessments at three, six, and twelve months post-injury; (2) an evaluation of sociodemographic, premorbid, and injury-related factors, and functional recovery contributing to worsening or improving outcomes after TBI. Using patient-reported outcome measures, recuperation trends after TBI were identified by applying Multivariate Latent Class Mixed Models (MLCMM). Instruments were grouped into TBI-specific and generic health-related quality of life (HRQoL; QOLIBRI-OS, SF-12v2), and psychological and post-concussion symptoms (GAD-7, PHQ-9, PCL-5, RPQ). Multinomial logistic regressions were carried out to identify contributing factors. For both outcome sets, the four-class solution provided the best match between goodness of fit indices and meaningful clinical interpretability. Both models revealed similar trajectory classes: stable good health status (HRQoL: *n* = 1944; symptoms: *n* = 1963), persistent health impairments (HRQoL: *n* = 442; symptoms: *n* = 179), improving health status (HRQoL: *n* = 83; symptoms: *n* = 243), and deteriorating health status (HRQoL: *n* = 86; symptoms: *n* = 170). Compared to individuals with stable good health status, the other groups were more likely to have a lower functional recovery status at three months after TBI (i.e., the GOSE), psychological problems, and a lower educational attainment. Outcome trajectories after TBI show clearly distinguishable patterns which are reproducible across different measures. Individuals characterized by persistent health impairments and deterioration require special attention and long-term clinical monitoring and therapy.

## 1. Introduction

Traumatic brain injury (TBI) remains one of the leading causes of death and disability in young adults [1,2,3] as well as in the elderly worldwide [4]. Individuals after TBI often experience profound physical, emotional, cognitive, and behavioral consequences that affect their ability to fully reintegrate into their work [5], family [6], and personal lives [7,8]. The financial impact on affected individuals, their families, and the health care system poses significant costs across Europe [9] and worldwide [10]. Therefore, it is important to identify factors related to the recovery process that may add prognostic value for patients after TBI of all severities. In particular, individuals who have experienced mild TBI could be more impaired than previously assumed [11,12,13]. Previous studies have shown that the time course of recovery varies across individuals, and that recovery rates may depend on the specific outcome, such as functional status [13], cognitive functioning [14], general health [15,16], and emotional status [17], as well as generic [18] and disease-specific health-related quality of life (HRQoL) [19,20]. Findings in this field of research may help to better inform patients and caregivers about treatment options and to tailor therapeutic and rehabilitative interventions accordingly.

Over the past three decades, several factors associated with recovery after TBI have been identified, including sociodemographic factors (e.g., age [21], education [14], sex, race [22,23,24], marital status, living situation [25], rural vs. urban residence [26]) and injury characteristics (e.g., injury severity [27], abnormalities in computed tomography scans [28,29], type of injury [30,31], length of coma [32], and length of post-traumatic amnesia [33,34,35]). Other clinical factors at the time of injury may also play an important role in the recovery process (e.g., functional independence at admission [36], presence of agitation or other pathognomonic signs [36]), as well as the nature of the injury and the presence of medical complications (e.g., skull fractures, hemorrhage, hematomas, intracranial pressure, midline shift [36]; intracranial bleedings [37]; hypoxia [38]; hypotension and acute trauma care treatments such as chemical paralysis or craniotomy, etc. [39]). Pre-morbid factors [40,41] such as pre-morbid IQ [42], history of psychological and psychiatric problems [43,44] and substance use disorders [43], prior acquired brain injuries [45], cognitive impairment or developmental disability [46], pre-injury employment status [30,47], and pre-injury occupation [34] can further influence the outcome post injury. Finally, the use of hospital and rehabilitation services (e.g., time from injury to rehabilitation admission [36], time spent in the intensive care unit [48], length of in-patient stay [34], length of hospitalization [37], discharge destination [49], rehabilitation intensity [50,51], and existence of health insurance [52,53]) can have an additional impact on the recovery process after TBI. 

The most common factors that are strongly associated with outcome after TBI are age [2,3,54,55], sex [56,57,58], education [14,59], premorbid health status [40,60], and TBI severity [57]. Some studies have found that younger individuals without pre-morbid health problems, and those with mild injuries, are more likely to show better recovery [61,62]. Sex is an independent predictor of TBI outcomes in acute hospitalization [63,64], inpatient rehabilitation [65,66], and long-term community settings [67]. While men are at higher risk of acquiring a TBI [57], most studies indicate that women are at greater risk of having less favorable outcomes compared to men [56,58].

Various studies have examined predictors of long-term outcomes after TBI [68,69,70]. In a recent US-based study [71], the following four latent profiles were derived from two-week assessments of neurobehavioral functioning: emotionally resilient, cognitively impaired, cognitively resilient, and neuropsychologically impaired. Inclusion of these classes in regression models substantially improved the prediction of the functional outcome, TBI-specific HRQoL, and symptom burden. In Europe, research has shown that individuals after TBI suffer from long-term functional and psychosocial adjustment difficulties that prevent them from fully re-integrating into daily life at pre-injury levels [72]. In addition, multiple studies have shown that affected individuals experience poor mental health [11,18,73,74,75] and reduced HRQoL [19,20,76,77]. Outcome after TBI is mostly assessed at fixed time points, but relatively little attention has been given to the outcome trajectory over time. Studies investigating the predictors of recovery of HRQoL, psychological, and psychosocial patient-reported outcomes (PROs) and their trajectories in individuals after TBI in Europe remain scarce. 

Therefore, the aims of the present study are the following:Identifying classes related to trajectories of improving or decreasing patient-reported TBI-specific and generic HRQoL and psychosocial and post-concussion symptom burden from three to twelve months after a TBI.Examining sociodemographic, premorbid, and injury-related factors associated with these recuperation classes of PROs.

Based on previous research findings, we expect that female gender, lower educational level, presence of premorbid psychiatric problems, as well as more severe TBI and injury severity are associated with less favorable outcome trajectories. 

## 2. Materials and Methods

### 2.1. Study Sample

Data were obtained from a European prospective, multi-center, longitudinal, cohort Collaborative European NeuroTrauma Effectiveness Research (CENTER-TBI; clinicaltrials.gov NCT02210221) study. Participants were recruited between 19 December 2014 and 17 December 2017 in 18 European countries and in Israel [78]. The inclusion criteria were as follows: clinical diagnosis of TBI; an indication for a computed tomography (CT) scan; admission to an emergency room with subsequent discharge (ER), or further admission to a hospital ward, or to an intensive care unit (ICU) within 24 h post-injury; and written informed consent. The core sample includes *N* = 4509 patients [79].

For our analyses of outcome trajectories, only individuals aged 16 years and above who had participated at three, six, or twelve months after injury were included. The final sample consisted of *N* = 2555 cases. Data were drawn from the core 2.0 dataset using the Neurobot data access tool. A detailed description of the sample selection process is provided in Figure 1.

### 2.2. Ethical Approval

The CENTER-TBI study (EC grant 602150) was conducted in accordance with all relevant laws of the EU, and of the country in which the recruiting sites were located, including but not limited to, the relevant privacy and data protection laws and regulations (the “Privacy Law”), the relevant laws and regulations on the use of human materials, and all relevant guidance relating to clinical studies including, but not limited to, the ICH Harmonized Tripartite Guideline for Good Clinical Practice (CPMP/ICH/135/95) (“ICH GCP”) and the World Medical Association Declaration of Helsinki entitled “Ethical Principles for Medical Research Involving Human Subjects”. The informed consent of the patients and/or their legal representative/next of kin was obtained according to local legislation for all patients recruited in the Core Dataset of CENTER-TBI and documented in the e-CRF.

### 2.3. Sociodemographic, Premorbid and Injury-Related Characteristics

Sociodemographic characteristics including sex, age, marital, educational and vocational status, and living situation were determined at the time of study enrollment.

Premorbid physical health status was assessed using the American Society of Anesthesiologists Physical Status Classification System (ASA) (i.e., normal healthy, mild disease, severe disease) [80]. Additionally, information on prior psychological problems, history of TBI, and developmental problems was collected based on self-reports. 

The Glasgow Outcome Scale—Extended (GOSE) [81] was used to rate the individuals’ functional status by a clinician using an eight-point scale: 1 (dead), 2 (vegetative state), 3 (lower severe disability), 4 (upper severe disability), 5 (lower moderate disability), 6 (upper moderate disability), 7 (lower good recovery), and 8 (upper good recovery). In addition to the clinical interview, its questionnaire version was also administered (i.e., the GOSE-Q [82]). The GOSE-Q was completed either by patients or by their proxies. To minimize information loss, missing GOSE values at three-, six-, and twelve-month assessments were substituted by the values derived from the GOSE-Q and clinical ratings in the centralized database of the CENTER-TBI study. Further details on the imputation procedure can be found elsewhere [83]. Since the GOSE-Q cannot distinguish between (2) vegetative state and (3) lower severe disability, these categories were merged into one. 

TBI severity was evaluated using the Glasgow Coma Scale (GCS) [84] which assesses level of consciousness following TBI. These data were combined with the presence of intracranial abnormalities (ICA) detected on a CT. Individuals were classified into four TBI severity groups according to the following cut-off values: uncomplicated mild (GCS ≥ 13, without ICA), complicated mild (GCS ≥ 13, with ICA), moderate (9 ≤ GCS ≤ 12), and severe TBI (GCS ≤ 8). 

Other TBI-related factors included the cause of injury (road traffic incident, incidental fall, violence/assault, other), clinical care pathways (emergency room (ER), admission, intensive care unit (ICU)), the length of the hospital stay (in days), the Injury Severity Score (ISS) [85], and the Brain Injury Score using the Abbreviated Injury Scale (AIS) [86].

### 2.4. Patient-Reported Outcome Measures 

The patient-reported outcome measures (PROMs) administered in this study capture TBI-specific HRQoL (QOLIBRI/-OS) and generic HRQoL (SF-12v2/-36v2), as well as several neuropsychiatric symptoms, including anxiety (GAD-7), depression (PHQ-9), post-traumatic stress disorder (PCL-5), and post-concussion symptoms (RPQ).

The Quality of Life After Traumatic Brain Injury—Overall Scale (QOLIBRI-OS) [87] measures TBI-specific HRQoL with six items rated on a five-point Likert-type scale ranging from ‘Not at all’ (1) to ‘Very’ (5) covering the following domains: ‘Cognition’, ‘Daily life and autonomy’, ‘Social relationships’, ‘Emotions’, ‘Physical problems’, as well as satisfaction with current situation and future prospects. An index score can be derived from the QOLIBRI-OS measure [88], ranging from 0 to 100, with higher values indicating higher HRQoL. A score below 52 indicates impaired HRQoL [89]. 

The Short Form—12 Health Survey version 2 (SF-12v2) [90,91] evaluates generic HRQoL. It comprises twelve items using different Likert-type response scales forming two summary component scores: the physical component score (PCS) and the mental component score (MCS). Both scores range from 0 to 100 and can be transformed into T-values using normative data. A score below 40 indicates impaired HRQoL.

The Generalized Anxiety Disorder—7 (GAD-7) [92] questionnaire measures seven symptoms of generalized anxiety disorder on a four-point Likert scale from ‘Not at all’ (0) to ‘Nearly every day’ (3). The total score ranges from 0 to 27, with higher values indicating greater impairment. Values of 5, 10, and 15 indicate mild, moderate, and severe anxiety, respectively [92]. Recently, psychometric performance after TBI and measurement invariance across different groups have been demonstrated in the CENTER-TBI sample [74,93].

The Patient Health Questionnaire—9 (PHQ-9) [94] contains nine items measuring major depression on a four-point Likert-type from ‘Not at all’ (0) to ‘Nearly every day’ (3). The total score ranges from 0 to 27, with values of 5, 10, and 15 indicating mild, moderate, and severe depression [94,95]. Like the GAD-7, the PHQ-9 has demonstrated satisfactory factorial structure, validity, and measurement invariance across several groups [74,93].

The Post-traumatic Stress Disorder Checklist for the DSM (PCL-5) [96] is a short screener for post-traumatic stress disorder (PTSD) covering 20 symptoms of PTSD according to the latest version of the Diagnostic and Statistical Manual of Mental Disorders (DSM-5) [97]. The instrument applies a five-point Likert-type scale from ‘Not at all’ (0) to ‘Extremely’ (4) with a total score ranging from 0 to 80. A cut-off score of 31 indicates clinically relevant impairment [98]. The PCL-5 and its translations have shown good psychometric properties and can be used as a reliable and valid screening measure of PTSD [99].

The Rivermead Post-Concussion Symptoms Questionnaire (RPQ) [100] comprises 16 items rated on a five-point Likert-type scale from ‘Not experienced at all’ (0) to ‘A severe problem’ (4). The items cover possible cognitive, somatic, and emotional symptoms after TBI. Individuals evaluate each symptom over the last seven days compared to the time before the injury. The total score ranges from 0 (no presence of symptoms) to 64 (most severe symptoms) with cut-offs of 13, 25, and 33 indicating mild, moderate, and severe symptoms, respectively [101], or a global cut-off with values greater than 12 indicating the presence of clinically relevant symptoms [101]. The RPQ and its translations have demonstrated good psychometric properties and can be used for screening post-concussion symptoms [99].

### 2.5. Data Analyses

#### 2.5.1. Multivariate Latent Class Mixed Models (MLCMM)

We applied Multivariate Latent Class Mixed Models (MLCMM) to detect the trajectories of outcomes in terms of ‘latent classes’ regarding the recuperation process after TBI. MLCMM is an extension of latent class mixture models using multiple dependent variables as indicators of an underlying, latent process. We grouped the outcomes into two categories: (1) TBI-specific and generic HRQoL (QoLIBRI-OS, SF-12v2 MCS, and SF-12v2 PCS) and (2) symptoms (PHQ-9, GAD-7, PCL-5, and RPQ). Using this method, participants with similar trajectories regarding all outcomes within each group are assigned to one class (e.g., participants with continuously high QoLIBRI-OS, SF-12v2 MCS, and SF-12v2 PCS levels to ‘stable good health’). MLCMM analyses were carried out using the R-package ‘lcmm’, version 1.9.2 [99,102], using the function ‘multlcmm’.

All models were run with 100 deviations from starting values and 10 iterations to identify a replicable Log-Likelihood maximum that was unlikely to be at a local maximum. The starting values of models with >1 class were derived from the 1-class-model. The best-fitting model was then fully fitted with a maximum of 500 iterations. For each of the two groups of outcomes, four statistical models with different parameter restrictions were processed: (1) random intercept and fixed slope mean for all members of a class; (2) random intercept and random slope mean for all members of a class; (3) random intercept and fixed slope mean for all members of a class as well as different intercept variances for each class; (4) random intercept and random slope mean for all members of a class as well as different intercept and slope variances for each class. Missing outcome data were handled by the MLCMM when at least one observation per outcome was available. 

Model selection was based on: (1) the Bayesian Information Criterion (BIC) [103] or sample-size adjusted BIC (SABIC), since the Akaike Information Criterion (AIC) often leads to an over-extraction of classes; (2) entropy, class size, and mean posterior probability for each class (minimum > 0.70) [104]; (3) the Lo-Mendell-Rubin adjusted likelihood ratio test (LMR-LRT), which compares a model with k classes with a model with k-1 classes [105]; (4) a visible, substantial improvement in fit indices (comparable to interpreting a scree plot in exploratory factor analysis) [106]. All models were run for increasing numbers of classes until non-convergence was reached or the fit indices started to increase again. To examine the stability of the class solutions, we repeated the MLCMM analyses within three subgroups with different levels of TBI severity: (1) uncomplicated mild, (2) complicated mild, and (3) moderate or severe TBI. 

#### 2.5.2. Multinomial Logistic Regressions (MLR)

For each of the two outcome groups (i.e., HRQoL, symptoms), multinomial logistic regressions (MLR) were applied to analyze contributing factors of class membership. These included sociodemographic (sex, age, level of education, employment and marital status, living situation), premorbid (health status prior to TBI, presence of psychological problems, TBI history), and injury-related (cause of injury, clinical care pathways, length of hospital stay, injury and TBI severity) variables and functional recovery status three months after TBI as measured by the GOSE. For the MLR, GOSE levels 7 (‘lower good recovery’) and 8 (‘upper good recovery’) were merged and used as a reference category, as differentiating them would result in numbers that would be too small for reliable statistical analysis. Missing covariate data were handled with the function ‘bootMice’ in the R-package ‘bootImpute’ [107]. First, the incomplete dataset was bootstrapped (*n* = 1000) and then multiple imputation (*n* = 5, with 20 iterations each) was applied using the R-package ‘mice’ [108]. MLR was conducted with all computed datasets (*n* = 5000) and the results were pooled using the function ‘bootImputeAnalyse’ of the ‘bootImpute’ package. All analyses were performed using R version 4.2. [109].

## 3. Results

### 3.1. Sample Characteristics

Detailed sociodemographic, premorbid, and injury-related information about the sample is presented in Table 1. These patient characteristics were also included in subsequent analyses aiming to identify their associations with outcome trajectories in terms of factors contributing to changes (i.e., worsening or improving) in health status after TBI.

#### 3.1.1. Sociodemographic Information

The mean age of the individuals included was 48.9 ± 19.5 years (Median = 50), covering an age range from 16 to 95 years. Approximately two thirds of the individuals were male (65.3%). More than one third (37.5%) had a higher secondary educational qualification. Nearly half of the individuals were employed full-time (44.7%), whereas a quarter of the sample (25.0%) were retired. Around half of the sample was married (44.5%) or living together with a partner (9.2%), and more than a fifth lived alone (20.6%).

#### 3.1.2. Premorbid and Injury-Related Information 

Most individuals had a healthy pre-injury health status (65.3%). In addition, 11.2% reported premorbid psychological problems, 10.1% reported a prior TBI, and 1.0% had developmental problems. 

For most of the individuals the cause of the injury was either a fall (44.5%) or a road traffic accident (41.0%). The clinical care pathway for nearly half of the sample was admission to an ICU (44.5%), admission to a hospital ward for 38.1%, and discharge after an ER visit for 20.7%. The mean length of hospital stay for those admitted either to the ward or the ICU was 11.4 ± 18.9 days (Median = 4.2). Almost 40% of participants sustained an uncomplicated mild TBI (39.7%) and more than one third a complicated mild TBI (35.2%), while 7.8% suffered from a moderate and 17.4% from a severe TBI. Three months after the TBI, nearly 60% had good recovery (58.7%), one quarter had moderate recovery (25.0%), and more than 15% had a severe disability (16.2%). For more details, see Table 1.

Compared with individuals excluded from the core sample of *N* = 4509, the sample used for the analysis had a lower percentage of male participants (*p* = 0.005), was more highly educated (*p* < 0.001), more likely to be employed full-time or part-time (*p* < 0.001), and was more likely to be married (*p* < 0.001). Additionally, the population analyzed was less likely to suffer from a severe disease (*p* < 0.001), was more likely to have been injured in a road traffic accident, and less likely to have had an incidental fall (*p* < 0.001). They were less likely to have been admitted to an ICU (*p* < 0.001), had a shorter hospital stay (*p* = 0.040), lower ISS (*p* < 0.001) and AIS scores (*p* < 0.001), a higher percentage of mild TBI (*p* < 0.001), and higher GOSE scores (*p* < 0.001). For more details, see Appendix B Table A1 and Table A2.

Table 2 provides descriptive statistics on PROMs assessed at three, six, and twelve months.

### 3.2. Classes of Outcome Trajectories

For both sets of outcomes (i.e., HRQoL and symptoms), the four-class-solutions with random intercepts and fixed slopes (model 1) represent the best match between statistical fit indicators and meaningful clinical interpretability [68]. Both models reveal similar groups in terms of trajectories: (1) stable good health status, in terms of a stable high TBI-specific and generic *HRQoL* (*n* = 1944, 76.1%) or a continuously low symptom level (*n* = 1963, 76.8%), (2) persistent health impairments, in terms of a continuously low TBI-specific and generic HRQoL (*n* = 442, 17.3%) or a continuously high symptom level (*n* = 179, 7.0%), (3) deteriorating health status, in terms of decreasing HRQoL (*n* = 86, 3.4%) or increasing symptoms (*n* = 170, 6.7%), and (4) improving health status, in terms of increasing HRQoL (*n* = 83, 3.2%) or decreasing symptoms (*n* = 243, 9.5%). For details on the results of the MLCMM analyses, see Appendix C (Table A3 and Table A4). Subgroup analyses at different levels of TBI severity (uncomplicated mild, complicated mild, and moderate/severe) generally supported the stability of the identified trajectory classes Appendix D (see Table A5, Table A6, Table A7, Table A8, Table A9, Table A10, Figure A1, Figure A2). However, the prevalence of trajectories differed significantly between the TBI severity groups for both outcome sets (all *p* < 0.001, except complicated mild vs. moderate/severe for symptoms: *p* = 0.079). Stable good health was the most frequent trajectory within all subgroups and for both outcome groups. Regarding HRQoL, change trajectories (i.e., improvement, deterioration) were the least frequent within all subgroups, but the ‘improving health’ class could be identified within the subgroup after complicated mild TBI. An increase in the frequency of unfavorable trajectories (i.e., deterioration or persistent impairment) was seen for symptoms (uncomplicated mild: 9.4%, complicated mild: 17.5%, moderate/severe: 17.7%) and, partly, for HRQoL (uncomplicated mild: 19.7%, complicated mild: 19.1%, moderate/severe: 25.8%).

Graphical plots of outcome trajectories for the four-class solutions are depicted in Figure 2 for all measures included in the model of the HRQoL outcome set as well as in Figure 3 in the model of the symptoms outcome set. The largest group with stable good health status showed high average TBI-specific (QOLIBRI-OS) and generic HRQoL (SF-12v2 PCS/MCS) as well as anxiety (GAD-7), depression (PHQ-9), PTSD (PCL-5), and post-concussion (RPQ) scores within the non-clinical range throughout the whole follow-up period. Within the group with persistent health impairments, unfavorable TBI-specific (QOLIBRI-OS) and generic HRQoL (SF-12v2) as well as moderate anxiety scores (GAD-7), PCL-5 levels above the screening threshold for PTSD, severe depressive (PHQ-9), and post-concussion symptoms (RPQ) were continuously observed. Persistent impairment occurred more frequently with respect to HRQoL (17.3%) than for symptoms (7.0%). Deteriorating health status was characterized by a marked decrease of high-level disease-specific (QOLIBRI-OS) and average generic HRQoL (SF-12v2) to lower scores than the class with persistent health impairments as well as an increase of the GAD-7, PHQ-9, PCL-5, and RPQ symptoms from subclinical or mild levels to moderate and clinically relevant levels. Improving health was marked by an increase in QOLIBRI-OS and SF-12v2 scores from the levels of persistent health impairment to stable good health as well as a decrease of the GAD-7, PHQ-9, PCL-5, and RPQ scores from moderate to subclinical levels slightly above the group with stable good health. 

HRQoL and symptom trajectory classes were associated significantly (χ^2^(9) = 1152.96, *p* < 0.001; Cramer’s *v* = 0.39), i.e., more favorable HRQoL trajectories generally corresponded to more favorable symptom trajectories and vice versa. Opposing change trajectories (e.g., improving HRQoL and deteriorating symptoms) rarely coincided. While participants with persistent HRQoL impairments exhibited varied symptom trajectories (all classes between 14.9% and 31.7%), the vast majority of participants with persistent impairments regarding symptoms also reported persistent HRQoL impairments (73.7%). For more details, see Appendix E (Figure A3), Table A1 and Table A2.

### 3.3. Association of Sociodemographic, Premorbid, and Injury-Related Factors with Trajectory Classes

#### 3.3.1. Results of Univariate Comparisons between Trajectory Classes 

Analyses of the sociodemographic characteristics revealed that the classes identified in both outcome groups (i.e., HRQoL and symptoms) differed significantly with respect to education level (*p*_Sympt_ < 0.001; *p*_HRQoL_ = 0.005) and employment status (*p*_Sympt_ < 0.001, *p*_HRQoL_ < 0.001). Significant age differences were only found for the four symptom-related classes (*p* < 0.001). Individuals with stable good health status were on average five years older compared to those with persistent health impairments. The four HRQoL-related classes differed significantly by sex, with approx. 12% more males in the stable good health status group compared to those with persistent health impairments (*p* < 0.001). Furthermore, individuals in the stable good health status group were more frequently single, less often divorced or separated (*p* < 0.001), and lived alone more frequently (*p* = 0.015) compared to those with persistent health impairments. 

In addition, both HRQoL and symptom classes differed significantly regarding several premorbid and injury-related characteristics: psychological problems, clinical pathways, length of hospital stay, injury severity score, brain injury severity score, TBI severity classification, and GOSE (all p_Sympt_ < 0.001, p_HRQoL_ < 0.001). The cause of injury only differed significantly for the symptom-related classes (*p* = 0.002). Here, individuals with stable good health status were less frequently (10%) victims of road traffic accidents and had more frequently (14%) sustained a fall compared to those with persistent health impairments. Physical health status prior to TBI only differed for the HRQoL-related classes, where individuals from the stable good health status group suffered less frequently from a mild or a moderate disease according to the ASA classification compared to those with persistent health impairments (*p* < 0.001). For more details on the descriptive statistics of the identified classes, see Appendix F (Table A11, Table A12, Table A13, Table A14).

The additional descriptive examination of the GOSE levels over three, six, and twelve months for each HRQoL and symptom class (see Figure 4 and Figure 5) revealed no clear linear association. Whilst participants with stable good health for both HRQoL and symptoms predominantly showed ‘upper good recovery’ and an increase thereof over time, results concerning the other three trajectories were less clear. A similar trend was visible regarding improving health, although the prevalence of lower GOSE levels was markedly higher than for stable good health. A slight improvement in GOSE levels was also seen in participants with persistent impairments regarding symptoms and HRQoL.

#### 3.3.2. Results of Multinomial Logistic Regression Analyses 

Multinomial logistic regression was applied to identify factors associated with individual class memberships within the four-class models for both outcome sets. The class with stable good health was used as a reference group (see Figure 6 and Figure 7; Appendix A). 

The odds of belonging to the group with persistent health impairments gradually increased with lower GOSE scores (OR_Sympt_ = 5.10–18.42; OR_HRQoL_ = 4.23–27.24). Furthermore, lower GOSE scores were not only associated with higher odds of belonging to the group with deteriorating health status (OR_Sympt_ = 2.88–3.92; OR_HRQoL_ = 3.92 (only GOSE 2/3), but improving health status as well, with more pronounced effects concerning HRQoL (OR_Sympt_ = 5.07–7.29; OR_HRQoL_ = 5.60–24.32). 

The same applies to prior psychological problems for both outcome sets, with one caveat: whereas the presence of psychological problems was related to higher odds of not only persistent health impairments (OR_Sympt_ = 3.43; OR_HRQoL_ = 2.25), but also improving health (OR_Sympt_ = 2.20; OR_HRQoL_ = 2.57) for both outcomes, higher odds for deteriorating health were only found regarding symptoms (OR = 2.00). Additionally, prior severe disease increased the odds of persistent health impairments with regard to HRQoL (OR = 2.22).

Furthermore, clinical care pathways revealed differential predictive values depending on the class under consideration. With respect to symptom trajectories, ICU admission was associated with higher odds of deteriorating health status (OR = 2.45) and significantly lower odds for improving health (OR = 0.48) as compared to admission to ER followed by discharge. In contrast to this, the clinical care pathway was not associated with HRQoL trajectories. Additionally, a longer hospital stay was associated with lower chances of persistent health impairments regarding symptoms (OR = 0.99) and improving HRQoL (OR = 0.98). None of the other injury-related characteristics identified as significant factors in the prior group comparisons showed meaningful significant effects in the multinomial logistic regression models.

Regarding sociodemographic factors, differential associations for age and sex were found in the two sets of outcomes (i.e., HRQoL and symptoms). Higher age was associated with significantly lower odds of belonging to the group with persistent health impairments, but only regarding symptoms (OR = 0.96). Male sex was associated with lower odds of belonging to the group with improving health (OR = 0.50) and persistent health impairments (OR = 0.61) with respect to HRQoL. Yet, no significant associations with the level of symptoms were found. For HRQoL, higher odds of persistent health impairments were associated with lower education (OR = 1.53–2.26), being unemployed (OR = 1.87), and living alone (OR = 1.60). For symptom trajectories, only the lowest level of education was associated with higher odds of persistent health impairments (OR = 2.65). 

To summarize, functional status (GOSE) three months after TBI and prior psychological problems were found to be the most consistent significant factors across both outcome groups (i.e., HRQoL and symptoms), with the odds for membership consistently increasing with lower GOSE levels for some groups. In general, only a few of the other premorbid and injury-related factors were significantly associated with group membership in both models (i.e., prior psychological problems and clinical care pathways). Regarding TBI-specific and generic HRQoL, compared to other trajectory classes there was a higher number of significantly contributing factors, with higher odds for females, lower education, being unemployed, living alone, prior severe disease and psychological problems, as well as lower GOSE-scores. 

## 4. Discussion

This study examined factors contributing to the longitudinal recuperation process of individuals after TBI, exploring influential sociodemographic, premorbid, and injury-related variables as potential predictors of the diverse development of PRO trajectories. For this purpose, we conducted multidimensional analyses of latent classes regarding trajectories in HRQoL and neuropsychiatric and post-TBI symptom status from three to twelve months after TBI and examined the influence of factors derived from the previous research. Consideration of intercorrelations between the outcome sets is mandatory because of the interdependencies across domains [110]. 

### 4.1. Classes of Patient-Reported Outcome Trajectories

Evidence pointed towards four different classes of outcome trajectories in both models, which are related to two different types of outcome sets—TBI-specific and generic HRQoL and symptom burden status. Both models represent the best match between statistical fit indicators and meaningful clinical interpretability. The identified outcome trajectories within one year after TBI, as well as information on associated factors, may provide clinicians with a basis for further follow-up programs and developing person-centered interventions. The stability of the classes could generally be demonstrated for different TBI severity levels. Only in participants with complicated mild TBI was no class with improving HRQoL identified. 

Both outcome models include a stable good health class, characterized by persistently high HRQoL and low symptom levels, which represents the most populated group in both models with nearly 75% of individuals. Secondly, a somewhat complementary class with persistent health impairment with persistently high symptom levels and low HRQoL emerged. In addition to these ‘stable’ classes, two ‘changing’ classes with almost identical patterns of recuperation trajectories were identified in both models: one class with improving health status in terms of increasing HRQoL and decreasing symptoms; another class with deteriorating health status in terms of decreasing HRQoL and increasing symptoms. Similar class solutions have been reported elsewhere in the literature concerning outcome trajectories for depression and PTSD symptoms after TBI (i.e., low symptoms/resilience, delayed symptoms, recovery, and persistent symptoms) [68,73]. In another study [111] focusing on generic and disease-specific HRQoL, GOSE, as well as post-concussion symptoms, only two to three different trajectories were identified. This might be due to differences regarding the study population with a much smaller sample size (N = 100) and the exclusion of participants after moderate/severe TBI. Similar to our study, however, the majority reported stable good health regarding most outcomes. Our study provides evidence for the generalizability of these different courses of outcome trajectories not only for mental health (i.e., anxiety, depression, PTSD) and TBI-related symptoms but also for HRQoL. Most notably, the greater part of individuals in our study belonged to the stable good health status group across several outcomes after TBI in terms of stable high or average levels of HRQoL as well as continuously low levels of depression, anxiety, post-traumatic stress, and post-concussion symptoms. Nonetheless, there were some other patient groups reporting more health impairments, which may be especially relevant to focus on in the clinical context. We are now therefore going to concentrate on these individuals.

### 4.2. Unfavorable/Non-Stable Trajectory Classes, Associated Patient Characteristics, and Clinical Implications

#### 4.2.1. Persistent Health Impairments

Persistent health impairments in terms of continuously high depressive, anxiety, PTSD, and post-concussion symptoms were observed in only 7% of the sample, whereas impairments concerning TBI-specific and generic HRQoL were found in 17%. This is in line with previous studies that have demonstrated adverse long-term effects of TBI on mental health [73] and TBI-specific and generic HRQoL [19,20,76,77]. Even twelve months after a mild TBI, a substantial proportion of patients show functional impairments [112,113], elevated post-traumatic depressive symptoms [68], and reduced satisfaction with life [112]. While the percentage is comparable to previous studies with respect to symptom level [68,73], persistent impairments regarding HRQoL seem to be more common after TBI. 

Compared with the stable good health status group, functional status at three months after TBI was by far the strongest contributing factor for persistent impairments, with a marked increase of the odds with lower levels of functioning for both symptoms and HRQoL. This is in line with a study showing consistent associations of better functional recovery with lower symptom burden and better quality of life [114]. In individuals suffering from complicated mild TBI, functional impairments and mental health problems were found to have a bidirectional, longitudinal relationship [115], which may contribute to maintaining HRQoL- and symptom-related impairment. 

Contrary to previous studies highlighting severity of injury [27,30,31,116] and TBI severity [28,29,57,117], as well as pathways or characteristics of clinical care [34,36,37,48,49] as modifying factors of the recuperation process, none of these were associated with persistent impairment in both the HRQoL and symptom models, despite significant differences regarding most injury-related factors found in pairwise group comparisons concerning HRQoL (e.g., longer hospital stay, higher severity of injuries and TBI). Since TBI severity and functional impairments are strongly associated [113], the inclusion of functional status in the models may have weakened the associations of other injury-related variables with class membership.

Concerning premorbid factors, prior psychological problems emerged as another risk factor for persistent health impairments, which is consistent with previous studies [44,60,118]. This effect was more pronounced for symptom burden and seems to indicate a persistence of premorbid psychological problems in some of the individuals after TBI. 

Severe premorbid physical health status also contributed significantly to persistent impairments regarding HRQoL, which is in line with previous studies [60,62]. Several sociodemographic factors were associated with higher odds for persistent HRQoL impairments, such as female sex, a lower educational level, being unemployed, and living alone, many of which have been associated with TBI outcomes and recovery in previous studies (e.g., sex [56,57,58,63,64,65,66,119], education [14,30,59], pre-injury employment status [30,47]). Age, which has been identified as another significant factor in previous studies [2,3,54,55,61,62], was not associated with persistent HRQoL impairment, but was related to symptoms. Contrary to previous studies [62,120], younger age was not associated with better recovery, but with increased odds of persistent symptom-related impairments. The effect of younger age on the risk of developing persistent health impairments after a TBI may be explained by the fact that these individuals are more likely to suffer a TBI because of road traffic accidents. They are more likely to be admitted to an ICU due to a higher degree of complications because of the traumatic event (e.g., more severe extracranial injuries or a lower degree of functional recovery). 

Taken together, individuals with persistent health impairments are more likely to be female, younger, have lower levels of education, live alone, have preexisting physical and mental health problems, and have unfavorable functional recovery compared with individuals with stable good health status after TBI. 

Therefore, it is particularly important to focus on patients’ medical history and consider their sociodemographic characteristics when selecting appropriate follow-up programs and targeted interventions. For example, psychological treatment may stabilize health status after TBI and help to tailor interventions for the health and daily life consequences of TBI. However, individuals after TBI are a heterogeneous group and identifying appropriate interventions is challenging. The biopsychosocial framework may be useful in devising interventions that consider personal, comorbid, and injury-related factors. 

#### 4.2.2. Deteriorating Health Status

Much like in previous studies [68,73], a relatively small percentage of the sample displayed deteriorating health status, in terms of decreasing TBI-specific and generic HRQoL (3.4%) or increasing symptoms (6.7%). This group comprises pre-injury psychological comorbidity in addition to injury-related psychological distress, which may complicate a health condition and hamper differential diagnosis. Together with the ‘persistent impairment’ group, about 13 to 20% of the TBI patients, depending on the PROs, display significant impairments 12 months after injury and thus require long-term follow-up. However, it should be noted that there are other needs not captured by the PROMs used in this study (e.g., motor, cognitive, and psychological TBI sequelae, return-to-work issues, etc.). Therefore, special attention should be given to this vulnerable group to ensure that their needs are appropriately addressed. Early interventions may be required for detecting, preventing, and treating these individuals as old symptoms may reappear and new symptoms can arise after a trauma. Furthermore, it seems likely that fewer individuals show consistent patterns of change across several outcomes and that those with inconsistent or opposing changes were possibly classified into one of the ‘stable’ categories. This could perhaps be due to the use of our multivariate approach (MLCMM) which categorizes individuals with a similar pattern across several outcomes into one class. 

Similar to the group with persistent health impairments, functional status at three months after the injury significantly increased the odds of deteriorating health status as compared with the stable good health group for both symptoms and HRQoL, but the association was weaker and less systematic. Although this group initially showed low symptom severity, prior mental health problems were significantly associated with a deterioration in health, albeit this effect was weaker as compared to persistent health impairments. Prior studies have demonstrated that mental health problems may contribute to a delayed onset of depression [68] and post-concussion symptoms [60]. On the other hand, contrary to our results, in the study of Sigurdardottir et al. [73] no psychiatric problems were reported one year prior to the TBI in the group with ‘delayed’ symptoms, perhaps due to the small group size (*n* = 8). A delayed increase of post-concussion symptoms can be accompanied by an increase in depressive symptoms, as there is a certain overlap in symptoms (e.g., feeling depressed, difficulty concentrating, sleep disturbances). An interesting difference emerged regarding clinical care pathways. Admission to an ICU was a significant risk factor for deteriorating health regarding symptoms, but not regarding HRQoL. Again, all other injury-related characteristics were not significantly associated with group membership in the regression models, despite significant differences in pairwise comparisons (e.g., higher severity of injuries and TBI). 

In conclusion, interventions should consider premorbid health status to prevent later manifestation of symptom burden, especially in those patients with functional impairments, a history of psychological problems, and those who were treated in an ICU. Psychosocial factors (e. g., difficulties with work) not considered in the current study could perhaps be associated with the deterioration of health following TBI, emphasizing the need for psychological treatment and rehabilitation.

#### 4.2.3. Improving Health Status

In nearly 10% percent of individuals after TBI, we observed an improving health status in terms of decreasing depression, anxiety, and post-traumatic and post-concussion symptoms, which is, again, comparable to previous research [68,73]. However, only 3% of them recovered regarding TBI-specific and generic HRQoL. This could be because individuals from this group had already reported an average level of HRQoL three months after the TBI. Similar to persistent health impairments and deteriorating health status, higher odds for improvement were associated with good recovery (i.e., GOSE 7–8) at three months post-TBI. This effect was stronger than for the group with deteriorating health, but less pronounced than for the group with persistent health impairments. Another similarity between these groups is the significant contribution of premorbid mental health problems to the probability of belonging to a trajectory class, which is also in line with the study of Bombardier et al. [68]. In pairwise comparisons, most injury-related characteristics differed for the stable good health group, but not for the groups with persistent health impairments and deteriorating health status. However, apart from functional status, only admission to a hospital ward or ICU was significantly associated with group membership in the symptom model—but in contrast to the group with deteriorating health, the odds were reduced. With respect to TBI-specific and generic HRQoL, female gender was associated with higher odds for improving health, which is similar to the group with persistent health impairments. In contrast to the group with persistent impairments, however, education, employment status, and living situation did not predict improving health. Further studies are necessary to identify additional psychosocial factors that support recovery. Overall, it can be concluded that individuals who exhibit stable premorbid health conditions (e.g., absence of severe diseases or psychological problems) are more likely to have a favorable recovery after TBI. This again highlights the need for accurate diagnosis to guide the selection of further interventions. 

### 4.3. Strengths & Limitations

A core strength of our analysis is that we simultaneously investigated two different sets of outcomes, including multiple measures capturing symptom status as well as different instruments assessing TBI-specific and generic HRQoL. We performed a MLCMM analysis which so far, to the best of our knowledge, has never been used to investigate TBI recovery trajectories across all severity levels. 

Furthermore, recuperation trajectories for both sets of outcomes were estimated simultaneously across corresponding measures within each model. This was quite beneficial as our results revealed converging classes and courses of recovery trajectories within separate models related to different sets of outcomes. Applying latent class mixture models for a multitude of outcomes may lead to the spurious identification of many latent classes, complicating interpretation of the results. The chosen statistical approach (i.e., MLCMM), however, accounts for the intercorrelations of the single symptom- and HRQoL-related indicators. The majority reported high stable health with regard to HRQoL and symptoms, which may indicate a common underlying neurological and biological basis. Additionally, both HRQoL and symptom trajectories were moderately correlated, indicating that more favorable trajectories for one outcome tend to coincide with more favorable trajectories for the other outcome. However, whereas most participants with persistent impairment regarding symptoms also reported HRQoL impairments, the reverse was not the case, indicating that HRQoL is more strongly impacted by other aspects of the participants’ lives than only symptoms. A drawback of this approach is, however, that differential change patterns among separate measures of symptoms or HRQoL remain undetected. Because of this, it was not possible to test whether the neurological and biological basis of the symptoms and HRQoL measured by the different instruments is heterogeneous. However, this was beyond the scope of the present study and should be the subject of future research.

Another strength of this study was the large sample size including the full range of TBI severity. However, the current sample did consist of individuals who were eligible for outcome assessment throughout the year after the injury. Thus, our sample may be biased regarding an overrepresentation of individuals after mild TBI, which in turn reflects the distribution of severity in the overall TBI population. Based on the study design, individuals evaluated in the ER and then discharged did not participate in the twelve-month outcome assessments. Hence, we have no information about their outcomes, which may limit the generalizability of the findings. 

A further limitation may be seen in the fact that the analyses of trajectories of outcome after TBI were focused on PROs. Analyses of the trajectories of functional recovery, assessed by a clinical rating instrument, the GOSE and its questionnaire version, were therefore beyond the scope of this study. However, such an investigation could shed further light on the longitudinal development of outcome after TBI. In addition, the inclusion of further time points (e.g., two weeks and/or one year after TBI) to determine recuperation classes may allow outcome trajectories to be identified more accurately. A descriptive analysis of GOSE levels over time (three, six, and twelve months) for each trajectory class indicated that clinical and patient-reported ratings do not always concur. Some participants reported subjective impairments in spite of physical recovery and vice versa. A similar finding has already been reported when examining TBI-specific HRQoL in relation to functional recovery [121]. This underlines the importance of measuring patient-reported outcomes, since the GOSE does not capture every aspect of the participants’ lives. Further research into the latent classes identified in this study is encouraged to assess the needs of these patient groups in more detail and to derive more specific treatment and therapy options after TBI.

## 5. Conclusions

The analyses of outcome trajectories of individuals after TBI show clearly distinguishable patterns which are reproducible across different groups of outcomes (i.e., TBI-specific and generic HRQoL and symptoms) as well as within these outcome groups across different types of measures. The four classes comprise individuals with stable good health, persistent health impairments, improving health, and deteriorating health. Individuals after TBI in a particular latent class share a common singular pattern with respect to changes in both outcome sets. The factors contributing to outcome trajectories are consistent throughout all groups, including premorbid, injury-related, and demographic patient characteristics. Unfavorable outcome trajectories (i.e., persistent health impairments or deteriorating health status) are more likely to be present particularly in individuals suffering from premorbid physical and mental health problems. Thus, a differentiated medical history, including detailed psychiatric and physical anamneses, can support the identification of vulnerable patients. These findings can help health care providers, clinicians, and researchers to better understand the recuperation process and improve the treatment, care, and rehabilitation of patients. In addition, our results highlight that early treatment of psychiatric and functional impairments associated with TBI is crucial to prevent persistence, aggravation, and chronification of HRQoL impairments and symptoms. 

## Figures and Tables

**Figure 1 jcm-12-02246-f001:**
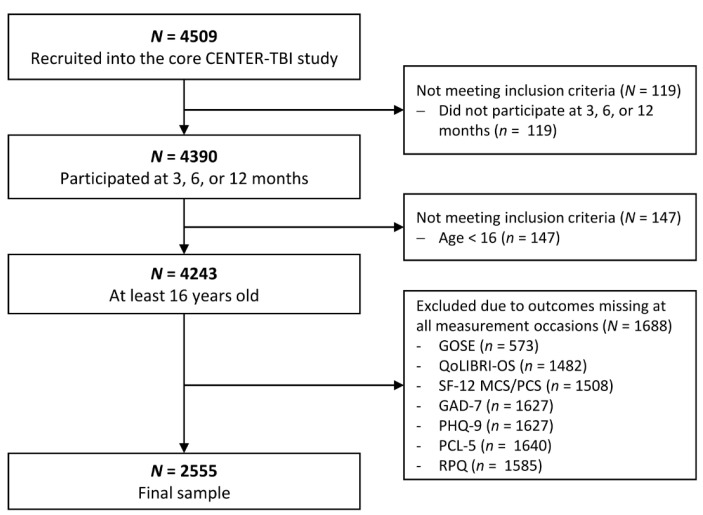
Flow chart of sample selection.

**Figure 2 jcm-12-02246-f002:**
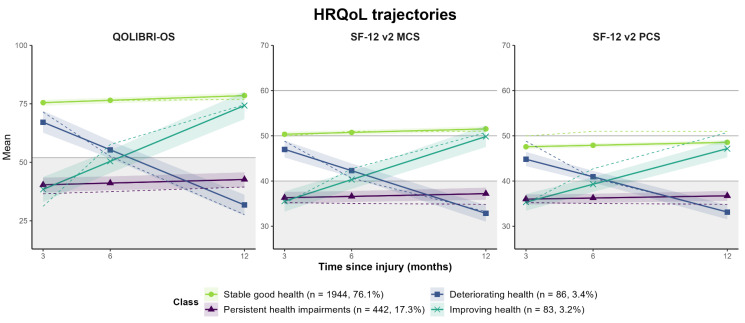
Generic and disease-specific HRQoL trajectories (QOLIBRI-OS, SF-12 v2 MCS/PCS) for the four-class solution. Solid, colored lines indicate predicted trajectories, dashed lines the observed values. Shaded areas around lines indicate 95% confidence intervals for predicted values. Gray lines mark cut-off for unfavorable values for the QOLIBRI-OS (<53) as well as mean (=50) and standard deviation (+/−10) for the SF-12v2 MCS/PCS. Gray area marks values below the cut-off for unfavorable levels of quality of life (QOLIBRI-OS < 52, SF-12v2 MCS/PCS < 50).

**Figure 3 jcm-12-02246-f003:**
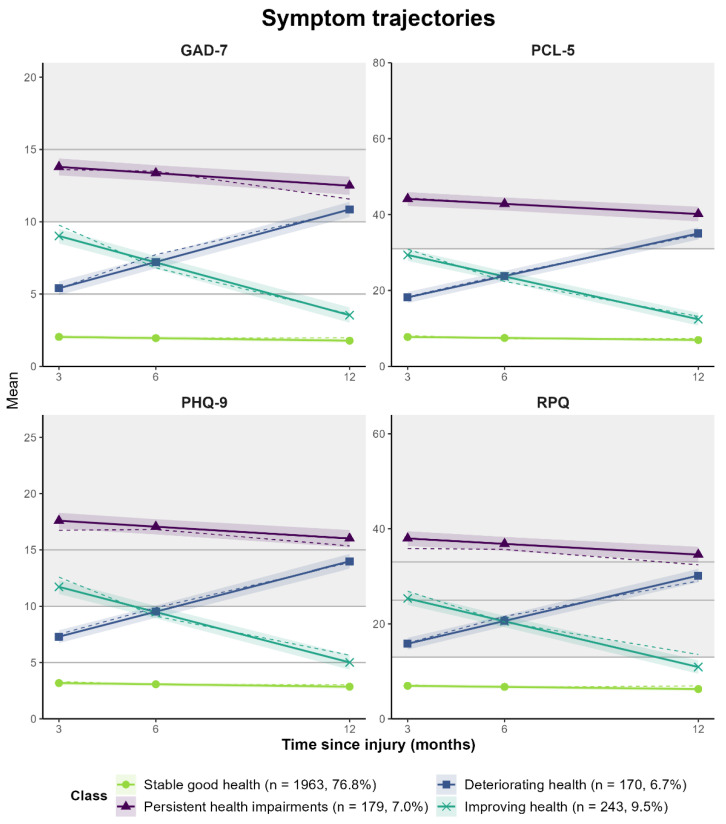
Symptom trajectories (GAD-7, PHQ-9, PCL-5 and RPQ) for the four-class solution. Solid, colored lines indicate predicted trajectories, dashed lines the observed values. Shaded areas around lines indicate 95% confidence intervals for predicted values. Gray lines mark cut-off values for categorical classification of outcomes (GAD-7: 5 = mild, 10 = moderate, 15 = severe; PHQ-9: 5 = mild, 10 = moderate, 15 = severe; RPQ: 13 = mild, 25 = moderate, 33 = severe; PCL-5 ≥ 31: PTBS screening). Gray areas mark values above the cut-offs for clinical relevance (GAD-7 ≥ 10; PHQ-9 ≥ 10; PCL-5 ≥ 31; RPQ ≥ 13).

**Figure 4 jcm-12-02246-f004:**
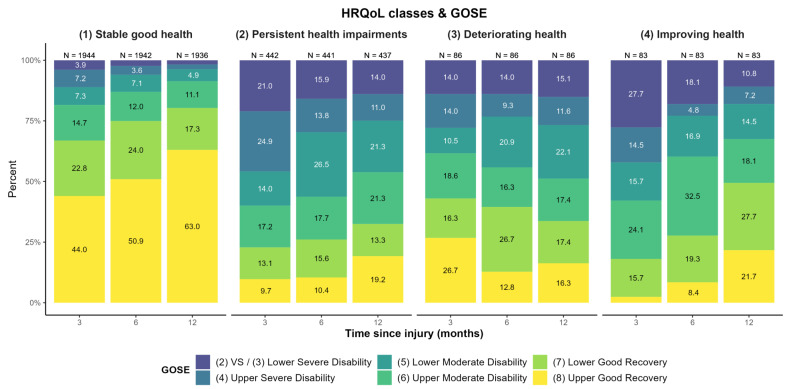
GOSE-levels at three, six, and twelve months after TBI for each generic and disease-specific HRQoL trajectory class. Decreasing sample sizes due to death of participants. Percentages < 3% not displayed.

**Figure 5 jcm-12-02246-f005:**
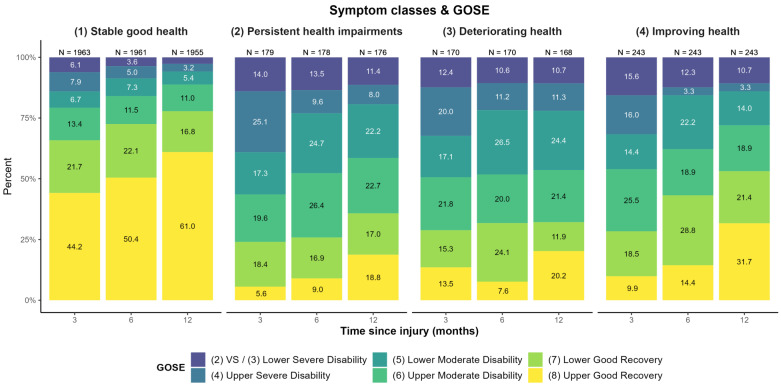
GOSE-levels at three, six, and twelve months after TBI for each symptom trajectory class. Decreasing sample sizes due to death of participants. Percentages < 3% not displayed.

**Figure 6 jcm-12-02246-f006:**
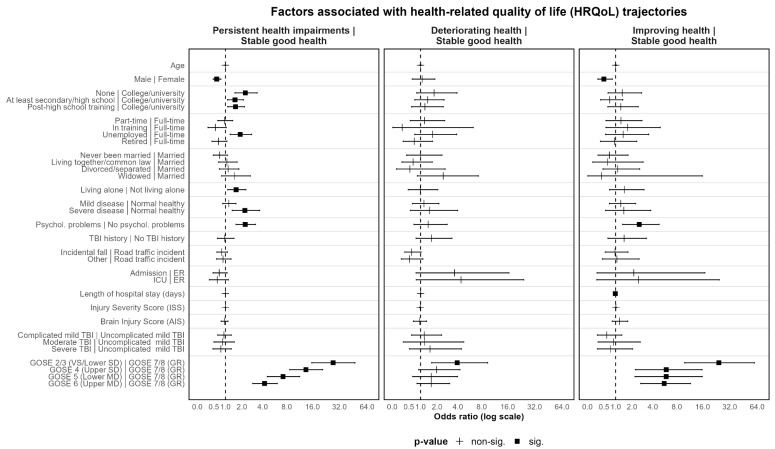
HRQoL trajectories (QOLIBRI-OS, SF-12 MCS, SF-12 PCS) for the four-class solution of model 1 (random intercept, fixed slope). Stable good health status class is used as the reference group. Odds ratios and 95%-confidence intervals depicted. Reference group for each categorical variable listed after vertical line. SD = severe disability, MD = moderate disability, GR = good recovery. Values below 1 indicate lower probability of belonging to the non-reference group (i.e., persistent health impairment, deteriorating health, or improving health) compared to the stable good health group.

**Figure 7 jcm-12-02246-f007:**
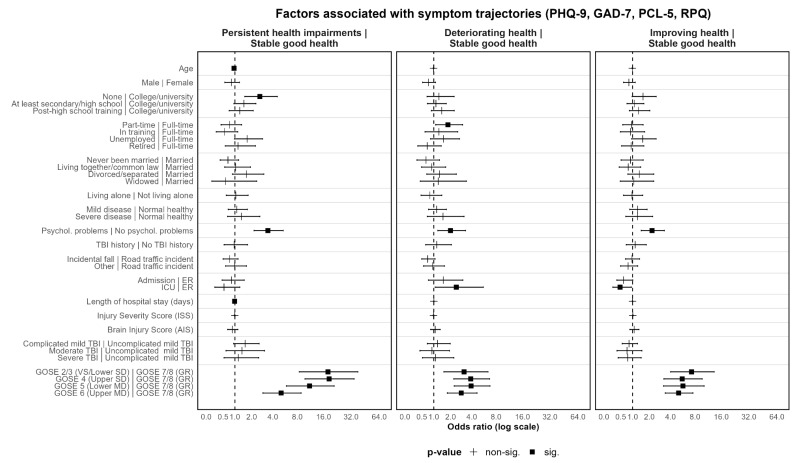
Symptom trajectories (PHQ-9, GAD-7, PCL-5, RPQ) for the four-class solution of model 1 (random intercept, fixed slope). Low stable health status class is used as the reference group. Odds ratios and 95%-confidence intervals depicted. Reference group for each categorical variable listed after vertical line. SD = severe disability, MD = moderate disability, GR = good recovery, ISS = Injury Severity Score, AIS = Brain Injury Score. Values below 1 indicate lower probability of belonging to the non-reference group (i.e., persistent health impairment, deteriorating health status, or improving health status) compared to the stable good health status group.

**Table 1 jcm-12-02246-t001:** Sociodemographic, premorbid, and injury-related characteristics of the sample (*N* = 2555).

			*N* (Valid %)	Missing, N (%)
Sociodemographic characteristics	Age (in years)	*M* (*SD*)	48.86 (19.49)	0 (0.0)
		Median	50.00	
		Range	16.00–95.00	
	Sex, *N* (%)	*Female*	886 (34.7%)	0 (0.0)
		Male	1669 (65.3%)	
	Education level, *N* (%)	None/primary school	327 (14.4)	281 (11.0)
		At least secondary/high school	852 (37.5)	
		Post-high school training	469 (20.6)	
		*College/university*	626 (27.5)	
	Employment status	*Full-time employed*	1072 (44.7)	159 (6.2)
		Part-time employed	269 (11.2)	
		In training	241 (10.1)	
		Unemployed	216 (9.0)	
		Retired	598 (25.0)	
	Marital status	Never been married	756 (31.3)	142 (5.6)
		*Married*	1073 (44.5)	
		Living together/common law	221 (9.2)	
		Divorced/separated	229 (9.5)	
		Widowed	134 (5.6)	
	Living alone	*No*	2026 (79.4)	3 (0.1)
		Yes	526 (20.6)	
Premorbid health status	Physical health Status (ASA) ^a^	*Normal healthy*	1510 (59.8)	31 (1.2)
		Mild disease	805 (31.9)	
		Severe disease	209 (8.3)	
	Psychological problems	*No*	2212 (87.9)	39 (1.5)
		Yes	304 (12.1)	
	TBI history	*No*	2190 (89.8)	116 (4.5)
		Yes	249 (10.2)	
	Developmental problems	*No*	2483 (99.0)	46 (1.8)
		Yes	26 (1.0)	
Injury-related factors	Cause of injury	*Road traffic incident*	1026 (41.0)	53 (2.1)
		Incidental fall	1113 (44.5)	
		Other ^a^	363 (14.5)	
	Clinical care pathways	*ER*	530 (20.7)	0 (0.0)
		Admission	974 (38.1)	
		ICU	1051 (41.1)	
	Length of hospital stay (days)	M (SD)	11.44 (18.86)	58 (2.3)
		Median	4.22	
		Range	0.00–370.50	
	Injury Severity Score (ISS)	*M* (*SD*)	18.62 (14.90)	33 (1.3)
		Median	14.00	
		Range	1.00–75.00	
	Brain Injury Score (AIS)	*M* (*SD*)	2.99 (1.35)	34 (1.3)
		Median	3.00	
		Range	0.00– 6.00	
	TBI severity	*Uncomplicated mild*	935 (39.7)	198 (7.7)
		Complicated mild	829 (35.2)	
		Moderate	183 (7.8)	
		Severe	410 (17.4)	
	GOSE (3 months)	Vegetative State/Lower Severe Disability	204 (8.0)	0 (0.0)
		Upper Severe Disability	274 (10.7)	
		Lower Moderate Disability	226 (8.9)	
		Upper Moderate Disability	398 (15.6)	
		*Lower Good Recovery*	*529 (20.7)*	
		*Upper Good Recovery*	*924 (36.2)*	

^a^ Category ‘Other’ includes ‘Other non-intentional injury’, ‘Violence/assault’, ‘Act of mass violence’, ‘Suicide attempt’, and ‘Other’. Note. Reference group of categorical predictors in *italic*. GOSE levels 7 (‘lower good recovery’) and 8 (‘upper good recovery’) were merged for the multinomial logistic regressions. *N* = absolute frequencies, % = relative frequencies, *M* = mean, *SD* = standard deviation, ER = emergency room, Admission = admission to a hospital ward, ICU = intensive care unit.

**Table 2 jcm-12-02246-t002:** Mean values for each outcome at the three time points.

		3 Months (*N* = 2309)	6 Months (*N* = 2281)	12 Months (*N* = 1802)	Number of Observations N (%)	
SF-12v2 MCS	*M* (*SD*)	47.05 (11.17)	47.72 (11.30)	47.44 (11.18)	1	518 (20.3%)
	Median	48.76	49.84	49.64	2	918 (35.9%)
	Range	9.71–72.86	7.56–71.98	10.16–73.80	3	1119 (43.8%)
	Missing	417 (16.5%)	367 (14.7%)	495 (25.4%)		
SF-12v2 PCS	*M* (*SD*)	43.65 (11.29)	45.73 (10.78)	46.43 (10.20)	1	506 (19.8%)
	Median	45.34	48.33	49.39	2	946 (37.0%)
	Range	11.63–69.52	9.86–65.11	12.28–65.49	3	1103 (43.2%)
	Missing	417 (16.5%)	367 (14.7%)	495 (25.4%)		
QoLIBRI-OS	*M* (*SD*)	67.39 (22.37)	68.62 (21.59)	68.27 (22.04)	1	506 (19.8%)
	Median	71.00	71.00	71.00	2	946 (37.0%)
	Range	0.00–100.00	0.00–100.00	0.00–100.00	3	1103 (43.2%)
	Missing	409 (16.1%)	373 (14.9%)	493 (25.3%)		
GAD-7	*M* (*SD*)	3.71 (4.57)	3.58 (4.49)	3.50 (4.38)	1	626 (24.5%)
	Median	2.00	2.00	2.00	2	903 (35.3%)
	Range	0.00–21.00	0.00–21.00	0.00–21.00	3	1026 (40.2%)
	Missing	524 (20.7%)	412 (16.5%)	540 (27.7%)		
PHQ-9	*M* (*SD*)	5.29 (5.40)	5.01 (5.31)	4.97 (5.38)	1	625 (24.5%)
	Median	4.00	3.00	3.00	2	905 (35.4%)
	Range	0.00–27.00	0.00–27.00	0.00–27.00	3	1025 (40.1%)
	Missing	519 (20.5%)	409 (16.3%)	548 (28.1%)		
PCL-5	*M* (*SD*)	13.12 (13.97)	12.20 (13.64)	12.30 (13.77)	1	632 (24.7%)
	Median	8.00	8.00	7.00	2	902 (35.3%)
	Range	0.00–79.00	0.00–80.00	0.00–79.00	3	1021 (40.0%)
	Missing	516 (20.4%)	414 (16.5%)	557 (28.6%)		
RPQ	*M* (*SD*)	11.29 (12.57)	10.96 (12.39)	11.00 (12.25)	1	587 (23.0%)
	Median	7.00	6.00	7.00	2	925 (36.2%)
	Range	0.00–61.00	0.00–64.00	0.00–57.00	3	1043 (40.8%)
	Missing	497 (19.6%)	391 (15.6%)	532 (27.3%)		

Note. N = absolute frequencies, % = relative frequencies, M = mean, SD = standard deviation.

## Data Availability

All relevant data are available upon request from CENTER-TBI, and the authors are not legally allowed to share it publicly. The authors confirm that they received no special access privileges to the data. CENTER-TBI is committed to data sharing and in particular to responsible further use of the data. Hereto, we have a data sharing statement in place: https://www.center-tbi.eu/data/sharing (accessed on 4 November 2021). The CENTER-TBI Management Committee, in collaboration with the General Assembly, established the Data Sharing policy and Publication and Authorship Guidelines to assure correct and appropriate use of the data as the dataset is hugely complex and requires help of experts from the Data Curation Team or Bio- Statistical Team for correct use. This means that we encourage researchers to contact the CENTER-TBI team for any research plans and the Data Curation Team for any help in appropriate use of the data, including sharing of scripts. Requests for data access can be submitted online: https://www.center-tbi.eu/data (accessed on 4 November 2021). The complete Manual for data access is also available online: https://www.center-tbi.eu/files/SOP-Manual-DAPR-2402020.pdf (accessed on 4 November 2021).

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
