# Peer review of "Impact of Sociodemographic, Premorbid, and Injury-Related Factors on Patient-Reported Outcome Trajectories after Traumatic Brain Injury (TBI)"

_jcm, 2023, doi:10.3390/jcm12062246_

Round 1

Reviewer 1 Report

This is a well-conducted study and an excellent report. My only comment pertains to the language in the Results. 

- Line 277: The majority (37.5%) had a higher secondary educational qualification. 

This is an incorrect statement. The largest number had a higher secondary educational qualification, but this does not make it the majority, which would be >50%. 

- Lines 277-278: Most of the individuals were full-time employed (44.7%)...

This is an incorrect statement. "Most" signifies >50%.

- Lines 279-280: Around half of the sample was married (44.5%) or living together with a partner (9.2%) and more than a fifth lived alone (20.6%).

This needs a comma after (9.2%).

- Lines 285-286: For most of the individuals the cause of the injury was a fall (44.5%) or a road traffic 285 accident (41.0%).

This is technically a correct statement but would read better if the word "either" was added (i.e.., "the cause of the injury was either a fall... or").

Author Response

Reviewer #1

This is a well-conducted study and an excellent report. My only comment pertains to the language in the Results. 

Response: Dear Reviewer, thank you very much for this encouraging feedback!

Comment 1:

- Line 277: The majority (37.5%) had a higher secondary educational qualification. 

This is an incorrect statement. The largest number had a higher secondary educational qualification, but this does not make it the majority, which would be >50%. 

Response: Thank you for your comments on the presentation of the descriptive results. We have adjusted the wording according to your suggestions. This was amended in the text in the following way:

Page 7, lines 279ff.

“More than one third (37.5%) had a higher secondary educational qualification.”

Comment 2:

- Lines 277-278: Most of the individuals were full-time employed (44.7%)...

This is an incorrect statement. "Most" signifies >50%.

Response: This was changed in the text in the following way:

Page 7, lines 280ff.

“Nearly half of the individuals were full-time employed (44.7%)…”

Comment 3:

- Lines 279-280: Around half of the sample was married (44.5%) or living together with a partner (9.2%) and more than a fifth lived alone (20.6%).

This needs a comma after (9.2%).

Response: A comma was added to this sentence according to your comment.

Comment 4:

- Lines 285-286: For most of the individuals the cause of the injury was a fall (44.5%) or a road traffic 285 accident (41.0%).

This is technically a correct statement but would read better if the word "either" was added (i.e.., "the cause of the injury was either a fall... or").

Response: Thank you! This sentence was changed according to your suggestion.

Reviewer 2 Report

The aim of the manuscript is to identify classes related to trajectory of an improvement or decrease of symptoms 3 to 12 months after a TBI, and to examine sociodemographic, premorbid and injury related factors.

They have a large group of patients representing mild to severe TBI. They used patient reports from several questionnaires and analysed the data with multivariate latent mixed models and multinomial logistic regressions. This resulted in four models and they identified persistent health impairments with need for long-term support. This is important and a promising result. However, there is a huge amount of data included in the analysis and it is sometimes hard to follow the discussion. Is it possible from the data to suggest more specified types of treatment for patients? How many in percentage are in need of long-term follow up?

Author Response

Reviewer #2

The aim of the manuscript is to identify classes related to trajectory of an improvement or decrease of symptoms 3 to 12 months after a TBI, and to examine sociodemographic, premorbid and injury related factors.

They have a large group of patients representing mild to severe TBI. They used patient reports from several questionnaires and analysed the data with multivariate latent mixed models and multinomial logistic regressions. This resulted in four models and they identified persistent health impairments with need for long-term support. This is important and a promising result.

Comment 1:

However, there is a huge amount of data included in the analysis and it is sometimes hard to follow the discussion.

Response: Thank you for your comment. We have revised the subheadings of the Results section and added subheadings to the Discussion section to further organize the content of these sections and better reflect the research questions. After identifying the trajectory classes (3.2.; page 9, lines 320 ff.), we proceeded with the factors associated with these classes (3.3.; page 12, lines 388 ff.). The results contain univariate comparisons between the identified trajectory classes, including additional analyses of functional status as assessed by the GOSE (3.3.1.; page 12, lines 390 ff.), as well as results of the multinomial logistic regression analyses (3.2.2.; page 13, lines 422 ff.). The discussion starts with the identified outcome trajectories (4.1.; page 19, lines 521 ff.) and moves on to the unfavourable or unstable trajectory classes, which were discussed in greater detail based on the univariate and regression analyses results and linked to the findings from the previous research (page 19, lines 556 ff.). We hope that our structure and our conclusions are now clearer and easier to follow.

Comment 2:

Is it possible from the data to suggest more specified types of treatment for patients?

Response: Thank you for this important comment. The purpose of the study was to identify classes of patients after TBI that could serve as a basis for further exploration of the special needs of these groups. Therefore, it was beyond the scope of this study to suggest more specific treatments. To do so, the groups should be studied in more detail, for example, by identifying comorbidities, the extent of extracranial injury, or by linking the classes to functional recovery status in more detail. We have presented some of these considerations in the Discussion (p. 22, lines 718 ff). We have added the following to the discussion to highlight the need for further investigation within the latent classes to draw conclusions for further treatments:

Page 23, lines 731ff.:

“Further research within the latent classes identified in this study is encouraged to assess the needs of these patient groups in more detail and to especially draw conclusions for more specific treatment and therapy for individuals after TBI.”

Comment 3:

How many in percentage are in need of long-term follow up?

Response: Thank you for this question. To make this clearer, we added the following sentence to the section describing the group with deteriorating health:

Page 21, lines 622 ff.:

“Together with the 'persistent impairment' group, about 13 to 20% of TBI patients, depending on the patient-reported outcome, show significant impairments 12 months after injury and thus require long-term follow-up.